# mRNA vaccination in people over 80 years of age induces strong humoral immune responses against SARS-CoV-2 with cross neutralization of P.1 Brazilian variant

Helen Parry[1], Gokhan Tut[1], Rachel Bruton[1], Sian Faustini[1], Christine Stephens[1], Philip Saunders[2], Christopher Bentley[1], Katherine Hilyard[3], Kevin Brown[4], Gayatri Amirthalingam[4], Sue Charlton[5], Stephanie Leung[5], Emily Chiplin[5], Naomi S Coombes[5], Kevin R Bewley[5], Elizabeth J Penn[5], Cathy Rowe[5], Ashley Otter[5], Rosie Watts[5], Silvia D'Arcangelo[5], Bassam Hallis[5], Andrew Makin[6], Alex Richter[1], Jianmin Zuo[1], Paul Moss[1]*

[1]Institute of Immunology and Immunotherapy, University of Birmingham, Birmingham, United Kingdom; [2]Clinical Lead, Quinton and Harborne PCN, Ridgacre House Surgery, Quinton, United Kingdom; [3]Vaccine Taskforce, Department for Business, Energy and Industrial Strategy, London, United Kingdom; [4]National infection Service, Public Health England, London, United Kingdom; [5]National infection Service, Public Health England, Porton Down, Salisbury, United Kingdom; [6]Oxford Immunotec Ltd, Abingdon, United Kingdom

*For correspondence:
p.moss@bham.ac.uk

**Abstract** Age is the major risk factor for mortality after SARS-CoV-2 infection and older people have received priority consideration for COVID-19 vaccination. However, vaccine responses are often suboptimal in this age group and few people over the age of 80 years were included in vaccine registration trials. We determined the serological and cellular response to spike protein in 100 people aged 80–96 years at 2 weeks after the second vaccination with the Pfizer BNT162b2 mRNA vaccine. Antibody responses were seen in every donor with high titers in 98 %. Spike-specific cellular immune responses were detectable in only 63 % and correlated with humoral response. Previous SARS-CoV-2 infection substantially increased antibody responses after one vaccine and antibody and cellular responses remained 28-fold and 3-fold higher, respectively, after dual vaccination. Post-vaccine sera mediated strong neutralization of live Victoria infection and although neutralization titers were reduced 14-fold against the P.1 variant first discovered in Brazil they remained largely effective. These data demonstrate that the mRNA vaccine platform delivers strong humoral immunity in people up to 96 years of age and retains broad efficacy against the P.1 variant of concern.

## Introduction

The current COVID-19 pandemic has led to over 2.6 million deaths but approval and widespread administration of several COVID-19 vaccine platforms have led to hope that the current pandemic may be brought under control. However, in order for this to be achieved, it will be essential that vaccine-induced immune responses are elicited effectively in people of older age (*Cox et al., 2020*). The functional quality of immune responses deteriorates with age and immunosenescence underlies

the increased burden of infectious disease in older people as well as impaired responses to vaccine challenge (*Ciabattini et al., 2018*; *Siegrist and Aspinall, 2009*). An exemplar is the efficacy of the annual inactivated influenza vaccine which is markedly suppressed in people ≥65 years (*Ciabattini et al., 2018*).

A range of potential mechanisms may underlie the development of immune senescence, including a reduction in the number of naïve T cells due to thymic involution and accumulation of memory cells, as well as an increased serum concentration of inflammatory molecules in a phenomenon termed inflammaging (*Egorov et al., 2018*; *Pietrobon et al., 2020*). Approaches such as higher antigen dose, adjuvant formulation, and usage of the live inactivated vaccination are being assessed to overcome these effects and improve vaccine efficacy. At the current time, there is insufficient evidence to assess the potential impact of immune senescence on response to the mRNA-based COVID-19 vaccines.

The nucleoside-modified RNA vaccine BNT162b2 from Pfizer BioNTech incorporating spike (S) is strongly immunogenic but participants over the age of 75 years comprised only 4 % of efficacy data (*Public Health England, 2021*). Furthermore, it is unclear if SARS-CoV-2 variants of concern (VOC) such as the P.1 variant which includes 10 mutations within the spike domain, including N501Y, E484K, and K417T, may mediate evasion of protective immunity (*Sabino et al., 2021*).

We undertook an analysis of serological and cellular immune responses to spike protein in 100 independently living people aged between 80 and 96 years who received BNT162b2 vaccine with a 3 -week interval between the first and second doses. We demonstrate strong humoral responses with evidence of broad neutralization of live Victoria virus and P.1 variants. Cellular immune responses were less marked and remained undetectable in 37 % of donors.

## Results

### Strong spike-specific antibody responses develop after BNT162b2 vaccination in older people

Serum samples were obtained from donors at 14–21 days following the second BNT162b2 vaccine (n=98). These were assessed initially for quantitative measurement of spike (S) responses using the Roche platform. S-specific responses were seen in all donors with 98 % above 1 in 50. Two donors had low but positive titers of 1 and 2.5 and antibody responses were confirmed on the MSD platform (*Figure 1A*).

Nucleocapsid (N)-specific antibody serostatus was used to determine if donors had evidence of prior natural infection with SARS-CoV-2. This was seen in 10 % of the cohort (10/98), with only 4 of these having reported previous symptoms indicating a high asymptomatic infection rate in this elderly cohort. Donors with prior SARS-CoV-2 infection had a median S-specific antibody titer of 32,250 which was 28-fold higher than the value of 1138 in those without prior infection (p<0.0001). No correlation was seen between the nucleocapsid titer and spike-specific titer amongst those with evidence of previous infection (*Figure 1B*).

Spike-specific ELISA (TBS) was also performed on eluates from dried blood spot (DBS) samples in order to assess the utility of the DBS platform in this age group (*Morley et al., 2020*). Very strong correlation was seen between values obtained from serum Roche anti-S ELISA and values from DBS eluates (r=0.68; p<0.0001) (*Figure 1C*). DBS was also obtained following the first vaccine administration in order to assess interim antibody generation and how this might predict values after the second vaccination. Antibody responses were detectable by DBS in 63 % of samples after the first vaccine (55/88) and this rose to 96 % when samples were taken after the second vaccine administration (91/95). Amongst those previously infected with SARS-CoV-2 only a 6 % rise in the antibody ratio was demonstrated between the first and second vaccines, suggesting minimal additional benefit from booster vaccination (*Figure 1D*). Amongst donors with no evidence of previous natural infection, a 3.1-fold increase in the ratio was observed between the first and second vaccines (median 1.2 [IQR 0.7–1.7] vs. 3.8 [IQR 2.8–4.7]; p=<0.0001 paired t-test [*Figure 1E*]).

Taken together, these results indicate strong spike-specific antibody responses develop in nearly all people aged above 80 years following the BNT162b2 vaccine regimen.

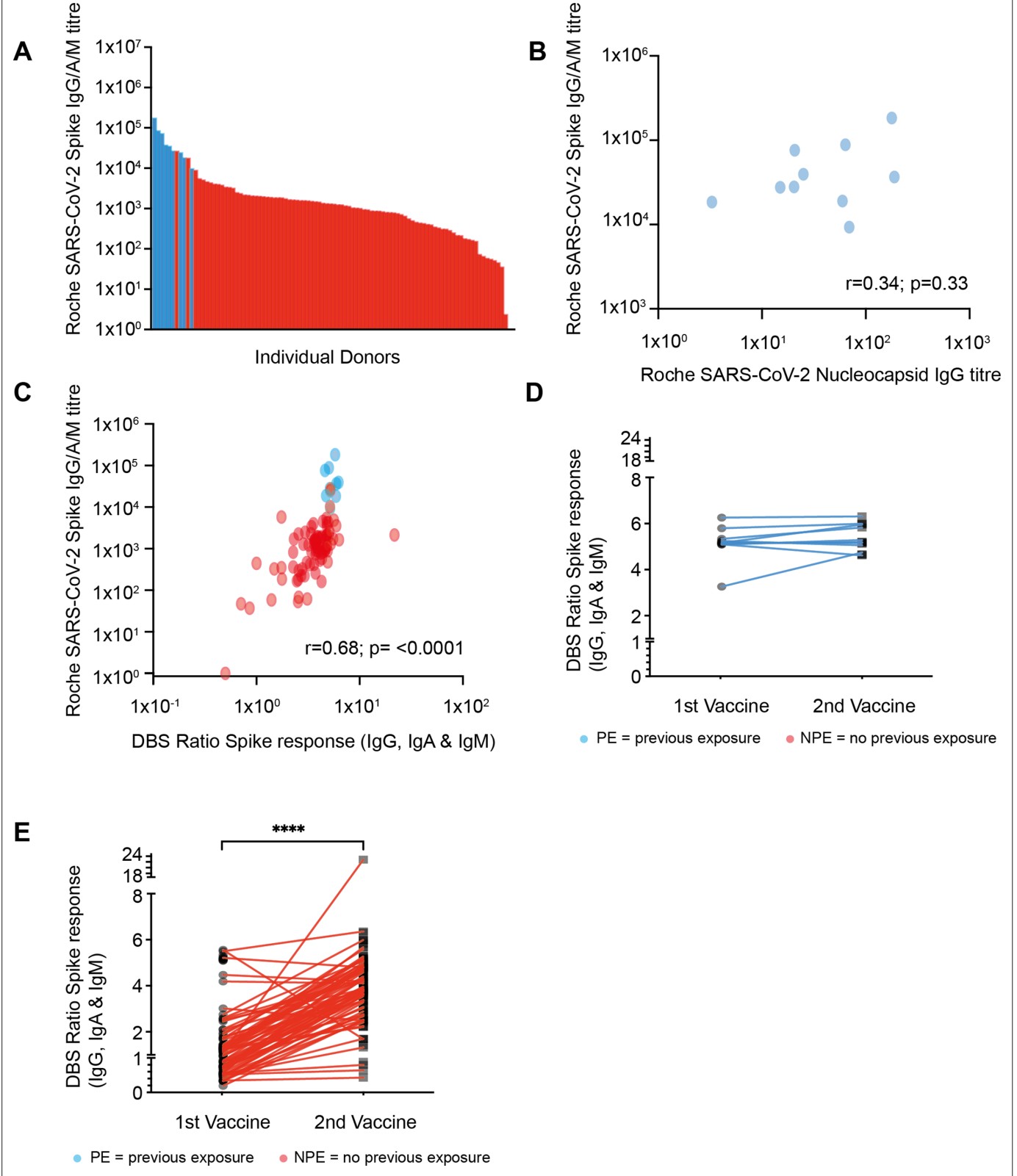

**Figure 1.** Strong antibody responses develop after vaccination with higher antibody levels seen in those with previous natural infection. (**A**) SARS-CoV-2 spike (S)-specific whole antibody titer after double vaccination. Blue bars represent participants where positive nucleocapsid (N)-specific serology indicates previous natural infection. (**B**) Comparison of S-specific and N-specific whole antibody titer after double vaccination amongst those with natural infection (R=0.34; p=033). (**C**) Comparison of S-specific whole antibody titer in serum with eluate ratio from dried blood spot (DBS). Blue dots

*Figure 1 continued on next page*

*Figure 1 continued*
represent participants where positive nucleocapsid (N)-specific serology indicates previous natural infection (r=0.68; p≤0.0001). (**D**) S-specific antibody response measured by DBS after the first and second vaccine dose amongst donors with evidence of natural infection. (**E**) S-specific antibody response by DBS after the first and second vaccine dose amongst donors with no evidence of previous natural infection (p≤0.0001).

The online version of this article includes the following source data and figure supplement(s) for figure 1:

**Source data 1.** Serum immunoglobulin isotype concentration in relation to spike-specific antibody response.

**Figure supplement 1.** Serum immunoglobulin isotype concentration in relation to spike-specific antibody response.

## Cellular responses are observed in 63% of participants following BNT162b2 vaccination and correlate with antibody response

Interferon gamma (IFN-γ) ELISpot analysis was then used to determine spike-specific T cell response following vaccination (n=98). Cellular responses against two peptide pools from the S1 and S2 spike domains, as well as membrane and nucleocapsid, were determined following overnight stimulation. Spike-specific T cell responses were detectable above threshold values in 63 % of participants (62/98) and responses against the S1 and S2 domains were equivalent (32 vs. 28 spots/million peripheral blood mononuclear cell [PBMC], respectively; p=0.35) (*Figure 2A*). The median magnitude of total spike-specific response was 84 spots/million PBMC (IQR 6–55) (*Figure 2B*).

Cellular responses were also assessed in relation to prior natural infection status. As expected, cellular responses against N were significantly greater in those with prior infection (N median 28 vs. 4 spots/million; p<0.0001). Interestingly, this group also developed threefold stronger T cell responses against spike peptides after vaccination (228 spots/million PBMC) compared to uninfected people (72 spots/million PBMC; p=0.0033) (*Figure 2C*). Similarly, an increased cellular response against the nucleocapsid and membrane proteins was also observed in people with previous natural infection and those who were uninfected (median N response of 12 spots/million for those with previous infection vs. 4 for those with infection naïve; p=0.0049 and for M response, amongst those previous exposed, a median of 28 spots/million compared to 4 p≤0.0001) (*Figure 2D and E*). Finally, we determined the relationship between the spike-specific antibody and T cell responses after vaccination and found these to be correlated (r=0.46; p=0.000003) (*Figure 2F*).

## Antibody and cellular responses following BNT162b2 vaccine are evident at extreme older age

Although all of our donors were over 80 years of age, we were interested to assess if age remained a determinant of vaccine response within older people. Although our cohort contained people up to 96 years of age no correlation between age and humoral or cellular responses were observed against the spike protein (*Figure 3*). As such these data demonstrate clearly that BNT162b2 vaccination elicits robust immunity even at extreme older age.

## Post-vaccination sera neutralize the Victoria strain and P.1 variant of concern

Aging can be associated with normal antibody levels but reduced functional activity and as such we next undertook live virus neutralization assays. Serial dilutions were performed to determine the reciprocal of the dilution that mediated 50 % neutralization (ND50) with the Victoria and P.1 variant of concern (n=20; *Figure 4A*). Assays were performed on 20 donors without evidence of prior infection and samples were selected to include donors with a range of magnitude of spike-specific response. Median ND50 was 2578 against the Victoria strain with almost all samples falling between 1000 and 15,895 (n=23). ND50 values for neutralization of the P.1 variant were reduced by 14-fold to a median value of 180 but remained above 36 for all donors (*Figure 4B*). As anticipated, overall anti-S IgG titer correlated strongly with neutralizing activity (Victoria r=0.857, p<0.0001; Brazil r=0.796, p<0.0001) (*Figure 4C*).

## Discussion

Older people comprise a large proportion of the population in many countries and it is essential that COVID-19 vaccines provide protection in this vulnerable group. The potential importance of immune

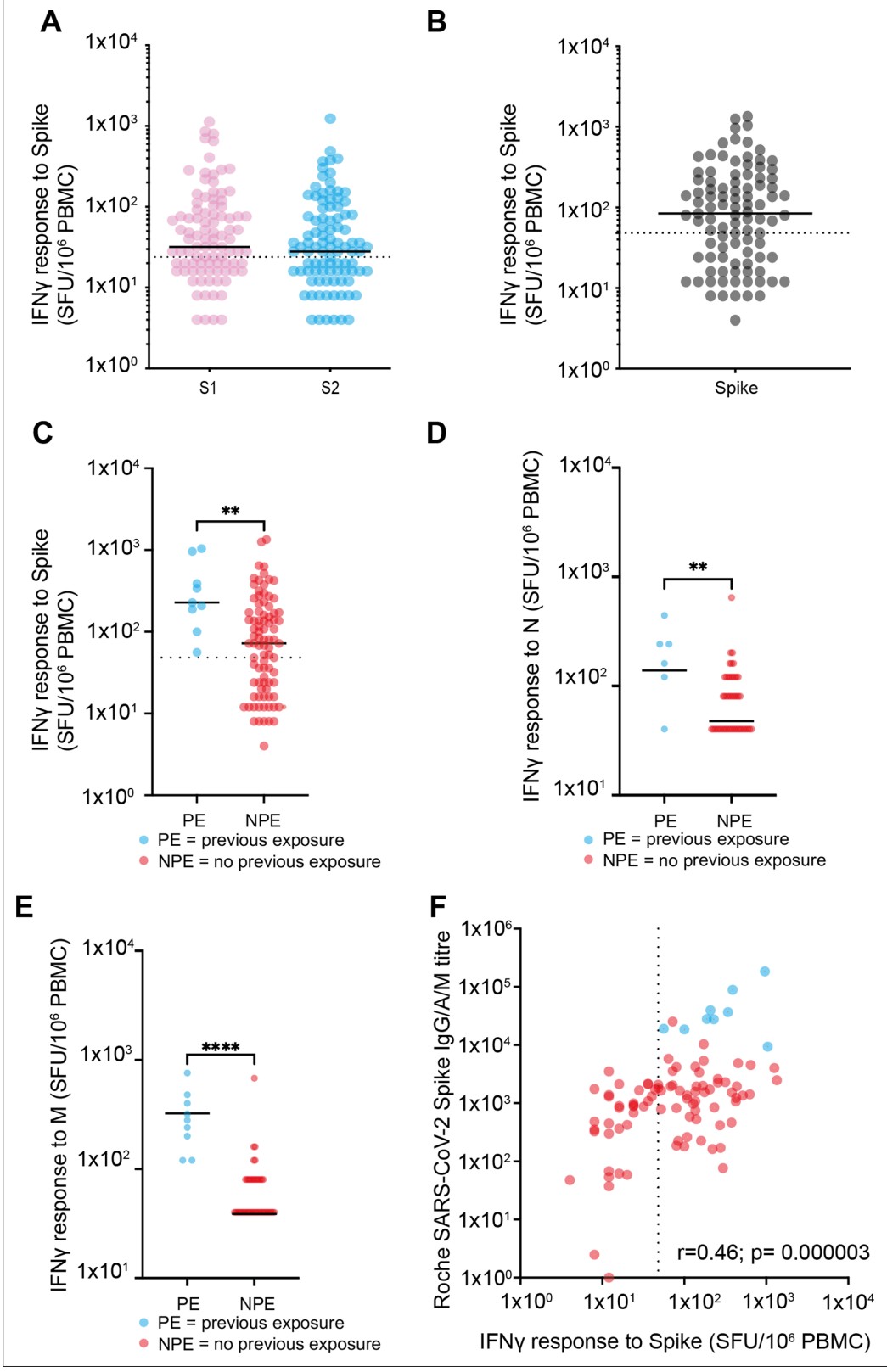

**Figure 2.** Spike-specific T cell responses after vaccination. (**A**) T cell responses against S1 domain and S2 domain as defined by IFNγ ELISpot assay. Black solid line indicates the median value of 32 against S1 and 28 against S2. Dotted line indicates cutoff for a positive response of 24 spots/million PBMC (n=98) (p=0.35). (**B**) Total spike-specific T cell responses as defined by IFNγ ELISpot assay. Black solid line indicates the median value of 84.

*Figure 2 continued on next page*

*Figure 2 continued*

Positive response defined as 48 spots/million (n=98). (**C**) T cell responses against spike by IFNγ ELISpot assay in relation to the history of previous natural infection. Blue indicates previous exposure (PE) (median 228 spots/million PBMC) and red indicates no previous exposure (NPE) (median 72 spots/million PBMC). Black solid line indicates the median value (n=98). Dotted line indicates cutoff for positive response. Solid black line indicates the median (p=0.0033). (**D**) T cell response against the nucleocapsid domain measured by IFNγ ELISpot assay in relation to the history of previous natural infection (p=0.049). (**E**) T cell response against the membrane domain measured by IFNγ ELISpot assay in relation to the history of previous natural infection (p≤0.0001). (**F**) Relationship of spike-specific whole antibody response by ELISA and spike-specific cellular response by ELISpot. Blue indicates PE and red indicates NPE. Dotted line indicates cutoff for ELISpot (r=0.46; p=0.000003). PBMC, peripheral blood mononuclear cell.

The online version of this article includes the following source data for figure 2:

**Source data 1.** Spike-specific T cell responses after vaccination.

senescence in relation to covid vaccination remains uncertain although a negative influence of age has been seen in vaccinees aged up to 77 years (*Abu Jabal et al., 2021*).

We detected spike-specific antibody responses in all older people which typically exceeded those seen after natural infection (*Krutikov et al., 2021*). It is likely that these findings underlie the excellent clinical protection that is emerging from vaccine trials in this population with 94 % protection from symptomatic infection in a real-world setting and 89 % efficacy in people over 80 years of age (*Dagan et al., 2021*; *Bernal et al., 2021*). Poor antibody responses were seen in two donors (*Figure 1A*) and it will be important to assess potential correlates of suboptimal response within individuals. As initial assessment, we measured total serum immunoglobulin IgG, IgA, and IgM levels across the whole cohort but did not see any correlation with the magnitude of vaccine response (*Figure 1—figure supplement 1*). A striking feature was that antibody responses remained robust in donors up to 96 years of age, indicating that mRNA vaccine efficacy for induction of humoral responses appears essentially independent of age within this older population.

In contrast, the induction of spike-specific cellular responses was less complete. T cell responses were present in 63 % of people but were of relatively low magnitude. The S1 and S2 domains were equivalently immunodominant. However, it is noteworthy that undetectable or very low cellular responses were seen in 37 % of people. The relative importance of cellular immunity in mediating clinical protection or sustaining humoral immunity is currently uncertain but these data indicate that this should be monitored prospectively. Indeed, a strong correlation was seen between the magnitude of the cellular and humoral immunity as observed in natural infection (*Zuo et al., 2021*). It will be of interest to assess how suboptimal antibody or cellular responses relate to the minority of people who fail to gain complete clinical protection from symptomatic infection following vaccination. Furthermore, It will be important to contrast values with those seen in younger donors in order to assess the potential impact of immune senescence on vaccine response. Of note, Salvagno et al. reported spike-specific antibody responses of 1364 U/ml with the same assay, which is comparable, although somewhat higher, to the value of 1138 seen in our study (*Salvagno et al., 2021*).

Primary SARS-CoV-2 infection presents a high clinical risk in people of this age but we observed that 10 % of the cohort had evidence of prior infection. This was associated with substantially stronger humoral and cellular immunity after vaccination. Indeed, no participants with a history of previous natural infection had suboptimal cellular immunity compared to 44 % in uninfected donors. Interestingly, we also observed strong responses after only the first vaccine dose in this group, with no significant increment after the second dose when administered with a 3 -week interval. As such these findings are comparable with those seen in healthcare workers which have led to suggestions that a single vaccine delivery may be sufficient for those with prior natural infection (*Levi et al., 2021*; *Krammer et al., 2021*).

The optimal in vitro correlation of natural protection is assessment of live virus neutralization and strong neutralization of the Victoria strain correlated with global spike-specific response. The P.1 VOC contains a range of mutations within the spike domain including E484K which can mediate escape from recognition by some antibodies (*Sabino et al., 2021*). A pronounced 14-fold reduction in median neutralization titer was observed with this variant which is larger than a 6.7-fold decrease in 30 younger vaccinees using pseudo-neutralization assays (*Garcia-Beltran et al., 2021*) and a 3.8–4.8-fold

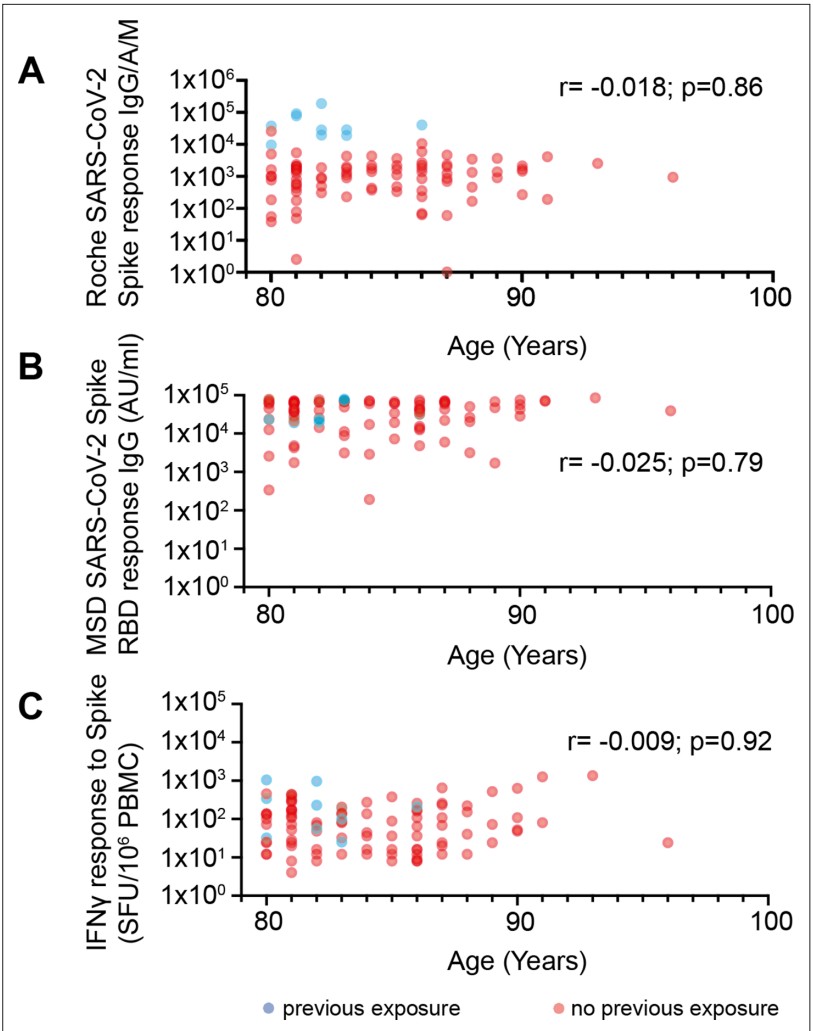

**Figure 3.** No correlation between age and vaccine response in donors 80–96 years of age. (**A**) Spike-specific whole antibody response using Roche ELISA in relation to age. Blue data points indicate previous exposure (PE) and red indicates no previous exposure (NPE) (r=–0.018; p=0.86). (**B**) RBD-specific IgG response (MSD) in relation to age. Blue data points indicate PE and red indicates NPE (r=–0.025; p=0.79). (**C**) Spike-specific cellular response by ELISpot in relation to age. Blue data points indicate PE and red indicates NPE (r=–0.009; p=0.92).

The online version of this article includes the following source data for figure 3:

**Source data 1.** Cellular immune response to vaccination.

decrease with the use of live virus (*Wang et al., 2021*). Further studies will be of interest to contrast these results with neutralization of other VOC (*Chen et al., 2021*). It is currently unclear how neutralization titers in vitro correlate with clinical protection but neutralizing activity remained somewhat robust across the cohort suggesting that vaccinated older people are unlikely to be highly susceptible to this rapidly emerging variant.

One of the limitations to this study includes the lack of pre-vaccination sample which is a reflection of the speed at which the vaccination program was rolled out in people over 80 years old and the challenge of operating within vaccine centers during national 'lockdown.' Future work should assess the longevity of the observed responses and neutralization of new variants that have emerged since the vaccination program started. This is now of great interest and may help to guide the need for further booster doses. Our work has focused solely on donors aged 80 years and older and, as such, it will also be important to see how immunity compares in younger cohorts who receive the BNT162b2 vaccine on a 3 -week interval.

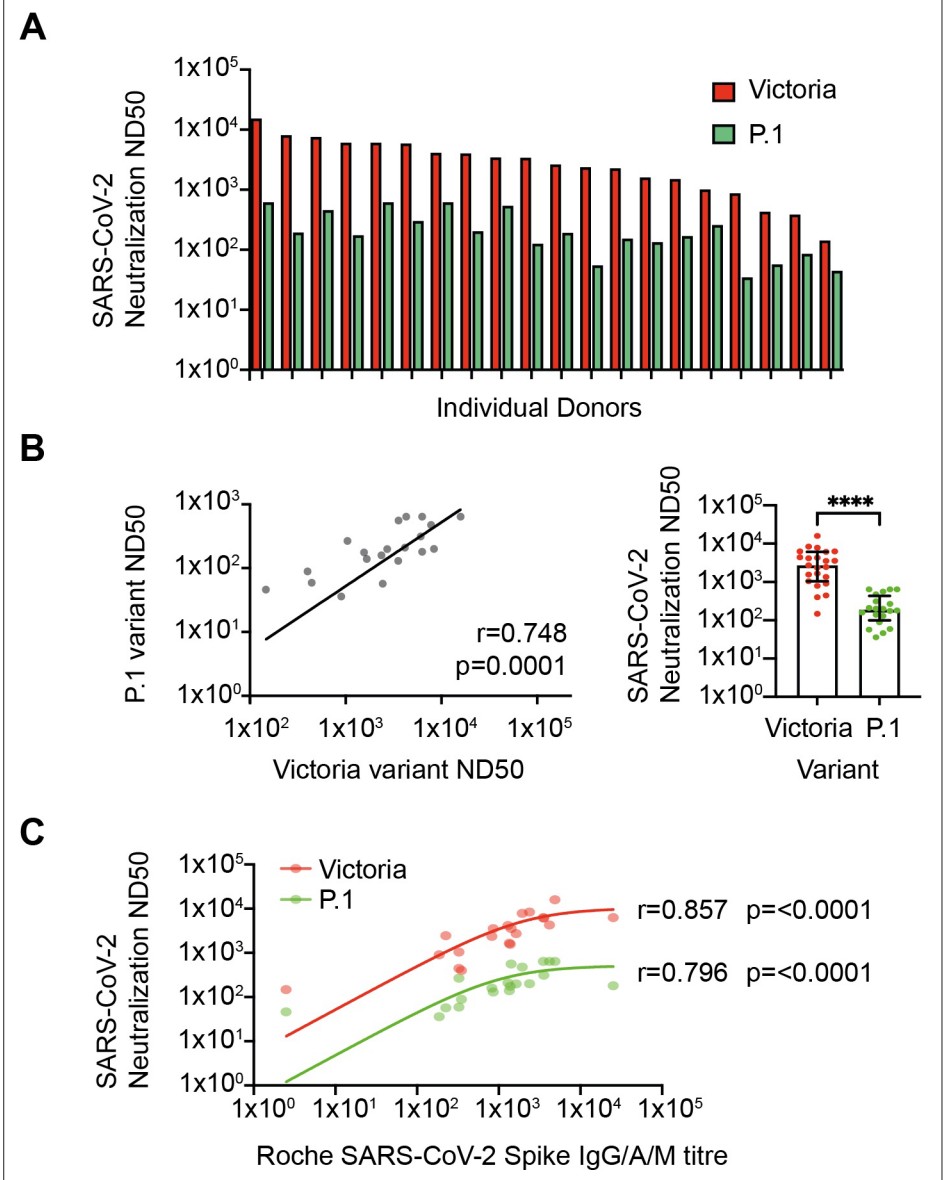

**Figure 4.** Neutralization of SARS-CoV-2 variants in vitro. (**A**) SARS-CoV-2 neutralization in vitro. Red bars represent neutralization against the Victoria variant and green bars show neutralization against the P.1 variant (n=20). (**B**) Relationship between neutralization of Victoria and P.1 variants (r=0.748; p=0.0001; n=20, respectively; p≤0.0001). (**C**) Correlation between spike-specific IgG titer and viral neutralization against Victoria and P.1 variants (r=0.857; p≤0.0001 and r=0.796; p≤0.0001, respectively).

The online version of this article includes the following source data for figure 4:

**Source data 1.** Neutralization of SARS-CoV-2 variants in vitro.

In conclusion, our findings demonstrate that the BNT162b2 vaccine generates robust humoral responses in older people which is likely to underpin the clinical efficacy of this regimen. High levels of spike-specific antibody should ensure control of the Brazilian variant of concern in most people despite a 14-fold drop in neutralizing activity. Further work will be of interest to define immune correlates that may be used to guide approaches to maintain immune responses in the longer term.

## Materials and methods

100 participants, aged 80 years or older, were recruited who were in living in community dwellings, able to attend a vaccination center, and did not require assistance with daily living or self-care. Written informed consent was obtained and the study was conducted according to the Declaration of Helsinki and good clinical practice. Participants were asked if they thought they had been infected with SARS-CoV-2 since the pandemic started.

The median age of participants was 84 years (IQR 80–87 or range 80–96) and 58 % were female. All participants received the Pfizer BNT162b2 COVID-19 vaccine with a 3 -week interval between the first and second doses. A phlebotomy sample was taken at 2 weeks following the second vaccine. A finger prick DBS sample was also taken at 3 weeks following the first vaccination in 88 donors and at 2 weeks following the second vaccination (*Morley et al., 2020*).

### Roche Elecsys electrochemiluminescence immunoassay (ECLIA)—IgG, A, and M assay against spike

Antibodies specific to SARS-CoV-2 were detected using electrochemiluminescence assays on the automated Roche Cobas e801 analyzers based at Public Health England (PHE) Porton. Calibration and quality control were performed as recommended by the manufacturer. Anti-nucleocapsid protein (NP) antibodies were detected using the qualitative Roche Elecsys Anti-SARS-CoV-2 ECLIA (COV2, Product code: 09203079190), whilst total IgG, A, and M anti-spike (S) antibodies directed at the receptor-binding domain were detected using the quantitative Roche Elecsys Anti-SARS-CoV-2 S ECLIA (COV2 S, Product code 09289275190), as previously described (*Manisty et al., 2021b*; *Manisty et al., 2021a*).

Anti-nucleocapsid results are expressed as cutoff index (COI) value, with a COI value of ≥1.0 considered positive for anti-nucleocapsid antibodies. Anti-spike results are expressed as units per ml (U/ml), with samples with a result of ≥0.8 U/ml considered positive for anti-spike antibodies within the fully quantitative range of the assay: 0.4–2500 U/ml. Samples >2500 U/ml were diluted further (1:10, 1:100, and 1:1000) to within the quantitative range.

### Mesoscale Discovery (MSD) IgG assay against spike and RBD

Quantitative IgG antibody titers were measured against spike (S) protein, nucleocapsid protein (N), and other antigens using the MSD V-PLEX COVID-19 Respiratory Panel 2 (96-well, 10 Spot Plate was coated with three SARS CoV-2 antigens (S, S-RBD S-NTD, and N)) (Cat # K15372U, Lot # Z0056764) from Meso Scale Diagnostics, Rockville, MD. Antigens were spotted at 200–400 µg/ml. Multiplex MSD assays were performed as per the instructions of the manufacturer. To measure IgG antibodies, 96-well plates were blocked with MSD Blocker A for 30 min. Following washing, with washing buffer, the samples were diluted 1:500 in diluent buffer. Reference standards and positive controls and diluted samples were added to the wells. After 2 hr of incubation and plates were washed 3 × with wash buffer and detection antibody (MSD SULFO-TAG Anti-Human IgG Antibody, 1/200) diluted in diluent 100 was added. After 1 hr of incubation at room temperature (RT), the plates were washed 3 × with wash buffer. MSD GOLD Read Buffer B was added and plates were read immediately using a MESO TM QuickPlex SQ 120. Text files were then generated from the Methodical Mind software then transferred to the MSD Discovery Workbench (v4.0) software. Data were then converted to AU/ml and exported as.csv files. The values from exported data were then adjusted for any sample dilution.

### Dried blood spot ELISA analysis—IgG, A, and M against trimeric spike

DBS analysis was carried out by Clinical Immunology Service (University of Birmingham). Capillary blood samples were collected on DBS cards (Ahlstrom Munksjo) from participants remotely and stored at RT. Samples were eluted in 250 µl of 0.05 % phosphate-buffered saline-Tween 20 (PBS, Oxoid Tween-20, Sigma-Aldrich) per blood spot and incubated overnight (12–16 hr) before centrifugation (10,600×*g* for 10 min). The DBS eluate was then applied to a pre-coated 96-well ELISA plate (The Binding Site, Birmingham, UK) containing stabilized trimeric SARS-CoV-2 spike glycoprotein and detecting IgG, IgA, and IgM antibody isotypes. The performance characteristics for this assay were assessed in 162 non-hospitalized mild to moderate disease PCR-positive individuals and 707 presumed COVID-19 negative samples from pre-2019. Sensitivity was 96.3 % (92.1–98.6) and specificity 99.3 % (98.4–99.8). The ELISA output result was reported as a ratio relative to the cutoff calibrator and multiplied by

the previously determined cutoff co-efficient to maintain batch-to-batch consistency, defined as 1.31 (*Cook et al., 2020*).

## Micro-neutralization assay

Neutralizing antibody titers were measured in heat-inactivated (56 °C for 30 min) serum samples. SARS-CoV-2 was diluted to a concentration of 1995 pfu/ml (150 ffu/50 µl) and mixed 50:50 in 1 % FCS/ MEM with doubling serum dilutions from 1:20 to 1:640 in a 96-well V-bottomed plate (samples were further diluted where appropriate to get ND50s into the working range of the assay). The plate was incubated at 37 °C in a humidified box for 1 hr to allow the antibody in the serum samples to bind the virus. Virus susceptible monolayers (Vero/E6 Cells) in 96-well plates were exposed to this serum/virus mixture. Plates were incubated in a sealed humified box for 1 hr before removal of the virus inoculum and replacement with semi-solid overlay (1% w/v CMC in complete media). The box was resealed and incubated for 20–24 hr prior to fixing for formaldehyde. Virus-specific foci were detected using a SARS-CoV-2 antibody specific for the SARS-CoV-2 RBD Spike protein and anti-rabbit HRP conjugate, infected foci were visualized using TrueBlueTM substrate. Stained foci were counted using Immuno-Spot S6 Ultra-V Analyzer (CTL) and resulting counts analyzed in SoftMax Pro v7.0 software. Values are stated as ND50, the reciprocal of the sample dilution causing 50 % of mock-neutralized virus control.

## Total immunoglobulin levels

Quantification of IgG, IgA, and IgM was evaluated using the COBAS 6000 (Roche) at the University of Birmingham Clinical Immunology Service. About 29 % of donors were found to have mild IgM deficiency (29/99), with 4 % deficiency in IgA (4/99), and 1 % in IgG (1/99).

## Cellular assays

PBMCs were isolated from a whole blood sample using 'T-Cell *Xtend*' (Oxford Immunotec) and Ficoll. After quantification and dilution of recovered cells, 250,000 PBMCs were plated into each well of a 'T-SPOT Discovery SARS-CoV-2' (Oxford Immunotec) Kit. The kit is designed to measure responses to four different but overlapping peptides pools to cover protein sequences of four different SARS-CoV-2 antigens, without HLA restriction, and includes negative and positive controls. Peptide sequences that showed high homology to endemic coronaviruses were removed from the sequences, but sequences that may have homology to SARS-CoV-1 were retained. Cells were incubated and interferon-γ secreting T cells were counted. A cutoff of 6+ spots per million on the S1 pool was defined as a positive result in line with the Oxford Immunotec diagnostic COVID Kit.

## Statistical analysis

Data were tested for normality using Kolmogorov-Smirnov analysis. For comparative analysis of antibody titers following the first and second vaccines, a paired t-test was performed. For comparison of S1 and S2 ELISpot responses, Wilcoxon test was performed and for unpaired analysis NPE versus PE and comparison of variant neutralisation, Mann-Whitney was performed. Spearman rank correlation was used for comparing assay platforms for titers and correlating T cell responses to IgG titer. Source data file available for all figures.

# Acknowledgements

This work was supported by the UK Coronavirus Immunology Consortium (UK-CIC) funded by DHSC/ UKRI and the National Core Studies Immunity program. The authors are very grateful for the support from patients and staff at Lordswood Medical Group and Ridgacre House Surgery. Jo McGlashan, Harriet Garlant, Bethany Hicks, Tom Coleman, Ann Varghese, Olivia Carr, Anaya Ellis, Caoimhe Kelly, Gabrielle Harker, Alexander Hargreaves, Sebastian Milward, and Stephen Taylor from PHE Porton Down kindly assisted with the development of the antibody and neutralization assays. This work was partially supported by the UK Coronavirus Immunology Consortium (UK-CIC) funded by DHSC/UKRI and the National Core Studies Immunity program.

# Additional information

## Competing interests

Andrew Makin: is affiliated with Oxford Immunotec Ltd. The author has no financial interests to declare.. The other authors declare that no competing interests exist.

## Funding

| Funder | Grant reference number | Author |
| --- | --- | --- |
| National Core Studies | Immunity programme | Paul Moss<br>Helen Parry<br>Gokhan Tut<br>Sian Faustini<br>Christine Stephens<br>Rachel Bruton |
| UK Coronavirus Immunology Consortium | UKRI/DHSC | Paul Moss<br>Helen Parry<br>Gokhan Tut<br>Sian Faustini<br>Christine Stephens<br>Rachel Bruton |

The funders had no role in study design, data collection and interpretation, or the decision to submit the work for publication.

## Author contributions

Helen Parry, Formal analysis, Investigation, Methodology, Writing - original draft; Gokhan Tut, Formal analysis, Methodology, Writing - review and editing; Rachel Bruton, Investigation, Methodology, Project administration, Writing - review and editing; Sian Faustini, Christine Stephens, Christopher Bentley, Sue Charlton, Stephanie Leung, Emily Chiplin, Naomi S Coombes, Kevin R Bewley, Elizabeth J Penn, Rosie Watts, Silvia D'Arcangelo, Investigation, Writing - review and editing; Philip Saunders, Ashley Otter, Alex Richter, Investigation, Methodology, Writing - review and editing; Katherine Hilyard, Kevin Brown, Gayatri Amirthalingam, Methodology, Writing - review and editing; Cathy Rowe, Methodology, Project administration, Writing - review and editing; Bassam Hallis, Andrew Makin, Investigation, Methodology, Validation, Writing - review and editing; Jianmin Zuo, Formal analysis, Investigation, Methodology, Writing - review and editing; Paul Moss, Funding acquisition, Project administration, Supervision, Writing - original draft

## Author ORCIDs

Stephanie Leung http://orcid.org/0000-0002-8880-2977
Paul Moss http://orcid.org/0000-0002-6895-1967

## Ethics

Human subjects: Informed consent, and consent to publish, was obtained. The study was approved by UPH IRAS ethics 282164, Health Research Authority UK.

## Decision letter and Author response

Decision letter https://doi.org/10.7554/eLife.69375.sa1
Author response https://doi.org/10.7554/eLife.69375.sa2

# Additional files

## Supplementary files

• Transparent reporting form

## Data availability

All primary data are available at https://doi.org/10.5281/zenodo.4740081.

The following dataset was generated:

| Author(s) | Year | Dataset title | Dataset URL | Database and Identifier |
|---|---|---|---|---|
| Parry H, Tut G, Zuo J, Moss P | 2021 | BNT162b2 vaccination in people over 80 years of age induces strong humoral immune responses with cross neutralisation of P.1 Brazilian variant | https://doi.org/10.5281/zenodo.4740081 | Zenodo, 10.5281/zenodo.4740081 |

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
