## [Decision Letter]

**Acceptance summary:**

The paper tackles an important topic, the immune response to mRNA PfizerCOVID19 vaccination in the oldest group of the population, the 80 to 96 years old. The authors show high SARS-CoV2 antibody responses and robust cellular responses in this age group at 2 weeks following the second vaccination, which did not decrease from 80 to 96 years of age. Moreover, the authors demonstrate the use of dry blood spots assays for these analysis. In addition, the antibodies also show neutralization against the P1 variant, even though decreased as compared to the original strain.

**Decision letter after peer review:**

Thank you for submitting your article "BNT162b2 vaccination in people over 80 years of age induces strong humoral immune responses with cross neutralisation of P.1 Brazilian variant" for consideration by *eLife*. Your article has been reviewed by 2 peer reviewers, and the evaluation has been overseen by a Reviewing Editor and Jos van der Meer as the Senior Editor. The following individual involved in review of your submission has agreed to reveal their identity: Debbie van Baarle (Reviewer #1).

The reviewers have discussed their reviews with one another, and the Senior Editor has drafted this to help you prepare a revised submission.

Essential revisions:

1. The absence of a pre-vaccination sample makes the visualization of the vaccine response complicated. Moreover, visualization of a robust vaccine response and low pre-vaccination titers would strengthen the data. Why did the authors not include a pre-vaccination sample? This is also of importance with respect to point 3, the use of many different antibody tests

2. The absence of comparison with a younger age group limits the use of the data presented in the paper. Even though robust antibody and cellular responses are shown in this age group, we don't know how that related to younger age groups and how 'impaired' immune responses are in this old age group. Do the authors have the possibility to compare their data to a younger age group? The data on cross-neutralization against the Brazilian variant is highly interesting. How does this difference in neutralization between the strains compares to younger age groups?

3. The authors use many different antibody tests to analyze their data. How do these compare? What is the rational of using all these different tests? Are these tests standardized and validated to golden standards? Why do the authors use a different test for the measurement of the serum and DBS antibodies? This makes the comparison of serum to DBS complicated and less fair. The use of so many different tests is confusing, as are their read outs; do they all measure IgG, or what are differences? Please provide a rationale for this per figure and explain differences more clearly. This aspect really affects clarity of the paper and needs additional attention.

4. The responses in the cellular tests are rather low with only 63% being positive. Which controls were included by the authors to investigate the robustness of the assay? These data are also rather limited by the lack of a pre-vaccination measurement. Do the authors have the possibility to compare to a pre-vac sample? As is done for the antibodies in Figure 1D. or to compare the results to a younger age group, to increase the reliability of the data shown.

5. We feel that the correlation in Figure 1B is statistically wrong, due to the inclusion of so many negatives for nucleocapsid. We would suggest to analyze this differently, by just comparing the spike response in 2 groups: the N positive and N negative ones. A correlation can be made only in the N positive ones.

6. Figure 1D: donors with previous exposure and no-previous exposure could be separated into two graphs to more clearly show that antibodies in those with prior exposure do not increase much after the second dose.

7. Line 267 – the authors mention that they examined cellular responses against spike, membrane and nucleocapsid but only the responses against spike are shown in Figure 2. Lines 274-275 – the cellular responses against N are only mentioned in text. These data should be shown as figures.

8. Why is the timepoint 2 weeks following vaccination chosen, and not 28 days post-vaccination as done in many clinical trials? How do the authors suggest that the responses do compare?

9. How were the 20 donors down-selected for the neutralization assays in Figure 4? Was this randomized?

10. Figure 4: it would be interesting to determine whether neutralization against a more neutralization-resistant variant (B.1.351 or B.1.617.2) is maintained in the elderly.

11. The correlation between age and the humoral and cellular response is difficult to interpret. Given the restricted age range, it can be expected that there will be limited differences, unless there are major differences in health between these individuals. The title of this paragraph should be tuned down and the restriction of this analysis should be mentioned in the Discussion section.

12. The Discussion section is somewhat limited. The authors should at least discuss the limitations of the study and indicate what additional information would be needed to draw firm conclusions.

13. Please rephrase the sentences 74 to 80 in the introduction, which lacks proper understanding of immunological ageing.

14. What do the authors mean by independently-living people as mentioned in sentence 87 in the introduction? Is that the proper terminology? Is community dwelling more appropriate?

15. Throughout the paper (including the title), the authors should avoid referring to the variants of concern with their country of origin. Either use the pango lineage names (P.1) or WHO's new Greek alphabet naming system for VOCs, in which P.1 is the γ variant.*Reviewer #1 (Recommendations for the authors):*

The authors tackle an important topic. It is important to analyze the immunogenicity of the mRNA vaccines in the oldest age group, the most vulnerable of the population. The notion of a robust immune response following vaccination is of importance for protection of this vulnerable age groups. Nevertheless, the authors did not compare the vaccine responsiveness to a younger age groups, to analyze the effect of age in these oldest olds. Moreover, the paper lacks a pre-vaccination sample. The authors show that the antibodies induced by the vaccination are also neutralizing against the Brazilian variant, a variant with a high rate of mutations in the spike protein, which is of importance to assess the potential threat of mutants on vaccine induced immunity. Nevertheless, the neutralization against the Brazilian variant is decreased as compared to the wild-type variant.

It would be important to investigate the long-term immune response to the vaccination in this age group, to investigate how long the oldest can be protected by the vaccines. The authors here only show the immune response at the very short-term, 2 weeks after 2nd vaccination, which does not provide answers for the long-term.

The authors provide an indication for the use of Dry blood spots for these kind of analysis and show a moderate correlation with the results found in blood. This analysis is however complicated by the use of different antibody assays. The authors found a moderate correlation between the cellular and antibody responses, but due to the high degree of variation in antibody assays used, this data is difficult the interpret.

Finally, the authors show a different vaccine response in the individuals that have been infected before vaccination. These participants were identified by the measurement of N specific antibodies, that are not induced in the vaccination. Spike specific antibody titers in this group were found substantially higher as compared to the only vaccinated group. A second vaccination in the previously infected group barely affected the antibody response, even in this oldest age group. Besides the limited number of participants, this information is important for guidance for the vaccination policies in the previously infected individuals. Also T cell responses were found enhanced in the previously infected group.*Reviewer #2 (Recommendations for the authors):*

Parry et al., examine anti-SARS-CoV-2 antibody and T cell responses following BNT162b2 vaccination in a cohort of 100 people aged 80-96 years. They showed that the majority of donors (98%) had spike-specific antibody responses but only 63% had detectable T cell responses against spike. These antibody and T cell responses did not correlate with age. Individuals who had prior infection had significantly higher antibody and T cell responses following vaccination. Finally, the authors showed that vaccinated individuals had strong neutralisation of a prototype strain of SARS-CoV-2, but reduced neutralisation against the P.1 variant of concern.

Strengths:

The authors have analysed antibody and T cell responses in a large group of people above the age of 80 (n=100), robustly showing that the elderly can respond to the Pfizer BNT162b2 vaccine and mount a strong antibody response. This data should instil confidence in the public that the elderly can mount a protective immune response against SARS-CoV-2 following vaccination with the Pfizer vaccine.

The authors also demonstrate that the strength of antibody and T cell responses did not correlate negatively with age (at least within the ages of 80-96).

Weaknesses:

While the authors achieved their aim in examining whether elderly individuals could mount an immune response against SARS-CoV-2 following vaccination, this paper lacked a control group of vaccinated younger individuals to make some valuable and interesting comparisons. With a group of younger individuals who received the Pfizer vaccine, the authors would be able to directly compare whether the elderly had weaker antibody and T cell responses following vaccination.

Opportunity:

The authors could also have analysed neutralisation against other variants of concern (especially the B.1.351 or B.1.617.2 variants that have been shown to be more neutralisation-resistant).

---

## [Author Response]

Essential revisions:1. The absence of a pre-vaccination sample makes the visualization of the vaccine response complicated. Moreover, visualization of a robust vaccine response and low pre-vaccination titers would strengthen the data. Why did the authors not include a pre-vaccination sample? This is also of importance with respect to point 3, the use of many different antibody tests

Thank you for your comment. We agree that it would have been useful to obtain a pre-vaccine sample but, as vaccination was approved rapidly in the UK, participants recruited to this study were some of the first in the world to receive the Pfizer-BioNTech vaccine outside of the registration trials. Vaccination centres were operating during a national lock down and study set up, ethical permission, and logistics for obtaining a phlebotomy sample (whilst maintaining social distancing and not hindering staff working to deliver the vaccines) were such that we were unable to obtain an initial pre-vaccine sample. We do, of course, use nucleocapsid-specific antibody assessment to define ‘pre-infected’ donors but the great majority of donors were infection-naïve and it would be expected that spike-antibody levels would have been undetectable in this cohort.

2. The absence of comparison with a younger age group limits the use of the data presented in the paper. Even though robust antibody and cellular responses are shown in this age group, we don't know how that related to younger age groups and how 'impaired' immune responses are in this old age group. Do the authors have the possibility to compare their data to a younger age group? The data on cross-neutralization against the Brazilian variant is highly interesting. How does this difference in neutralization between the strains compares to younger age groups?

Thank you. It would have been interesting to directly compare against younger donors but in the UK the standard 3-week interval vaccine was used only donors aged >80 years and health care workers. This is because, after these initial programmes, there was a mandatory increase in the time between doses. HCW are not an ideal control for this cohort as rates of viral exposure are likely to differ from our elderly cohort who have been ‘shielding’.

Salvagno et al., utilised the same assay platform as we describe and found, in health care workers with a median age of 44 years and without previous natural infection, a median titre of 1364 U/ml (761-2174) with 100% seropositivity following the second vaccine at day 50. This result is similar to our finding of 1138 U/ml following the second vaccine in those over 80. We have now added this to the list of references.

We agree that the data for neutralisation of P.1 are interesting. In addition to reference 20, we have also added a new reference from Wang et al., who report a 3.8-4.8-fold decrease in live virus neutralisation of this variant in younger donors.

3. The authors use many different antibody tests to analyze their data. How do these compare? What is the rational of using all these different tests? Are these tests standardized and validated to golden standards? Why do the authors use a different test for the measurement of the serum and DBS antibodies? This makes the comparison of serum to DBS complicated and less fair. The use of so many different tests is confusing, as are their read outs; do they all measure IgG, or what are differences? Please provide a rationale for this per figure and explain differences more clearly. This aspect really affects clarity of the paper and needs additional attention.

Thank you. We agree that the study used several platforms and, whilst one advantage of this is that it allows relative comparison between assays, it is important to justify why this was done. There were several reasons for this approach.

We utilised the Roche assay wherever a serum sample was available as this provided information on previous natural infection and also Roche is becoming an international standard test. The Roche assay provides IgG, A and M data against the receptor binding domain of Spike, rather than the total spike. This information has been added to the methods section. The Comparison data is available for Roche versus TBS DBS in figure 1C, with good correlation seen.

The MSD assay was used as this allowed us to assess the difference in IgG antibody response against spike and the receptor binding domain of the spike protein that is known to be important for neutralisation. This data was not available from the other platforms.

We were only able to obtain a pre-second vaccine sample by dried blood spot testing as phlebotomy bleed at this time point was not available as previously explained. The assay that has been developed for DBS testing is the Binding site assay which uses full trimeric spike and assesses the IgG, IgA and IgM response against Spike. The TBS DBS has been validated against WHO standard.

These different assays will allow other groups to assess the correlation between assays for their studies. To improve the clarity of what each assay reports, the subtitle for each assay in the methods section has been modified to incorporate this information.

4. The responses in the cellular tests are rather low with only 63% being positive. Which controls were included by the authors to investigate the robustness of the assay? These data are also rather limited by the lack of a pre-vaccination measurement. Do the authors have the possibility to compare to a pre-vac sample? As is done for the antibodies in Figure 1D. or to compare the results to a younger age group, to increase the reliability of the data shown.

Thank you for your comment.

The ELISpot assays were performed at Oxford Immunotec and is a standard assay that is widely used (https://www.tspotcovid.com) for diagnostic testing, vaccine assessment and research studies. As such we are confident in its performance although different cellular assays within laboratories will show variation in sensitivity.

It was not feasible to get a blood sample for PBMC extraction and cellular work at the pre-vaccination time point for the reasons stated in the answer to question 1. In order to obtain some information at the pre-second vaccine time point, we utilised DBS for Ab responses as this was performed as a self-testing kit. Unfortunately, DBS is only useful for serological assays and does not permit any cellular work.

Unfortunately we do not have a younger cohort who received 2 doses of Pfizer-BioNtech 3 weeks apart as this was not offered to younger ages in the UK and Oxford Immunotec do not hold any data for younger cohorts in this setting. However, we agree that the impact of immune senescence on cellular responses is a critical topic for future assessment.

5. We feel that the correlation in Figure 1B is statistically wrong, due to the inclusion of so many negatives for nucleocapsid. We would suggest to analyze this differently, by just comparing the spike response in 2 groups: the N positive and N negative ones. A correlation can be made only in the N positive ones.

Thank you. We have re-analysed the data as you suggest and now included an updated graph and correlation of those donors who are just nucleocapsid positive in Figure 1B. The correlation is indeed no longer present and we have changed the text accordingly.

6. Figure 1D: donors with previous exposure and no-previous exposure could be separated into two graphs to more clearly show that antibodies in those with prior exposure do not increase much after the second dose.

Thank you. We have separated the graphs and aligned them next to each other in figure 1D and 1E and the text has been amended accordingly.

7. Line 267 – the authors mention that they examined cellular responses against spike, membrane and nucleocapsid but only the responses against spike are shown in Figure 2. Lines 274-275 – the cellular responses against N are only mentioned in text. These data should be shown as figures.

Figure 2D and 2E have now been added to show the data for nucleocaspsid and membrane domain cellular responses in relation to previous natural infection and the text amended to incorporate these findings.

8. Why is the timepoint 2 weeks following vaccination chosen, and not 28 days post-vaccination as done in many clinical trials? How do the authors suggest that the responses do compare?

This time point was chosen to align with the original NEJM paper analysing the immunogenicity of the BNT162b2 vaccine from the phase 1 trial. Samples were taken at 7 and 14 days post the booster vaccine in this study which can be found at https://www.nejm.org/doi/full/10.1056/NEJMoa2027906. Based on the 30mg dose data in participants aged 65-85 years the geometric mean concentration of S1 binding IgG was 6014 U/ml whilst in our paper over those aged 80 and older, at the same time point was 1138 U/ml but this was utilising a different assay platform.

9. How were the 20 donors down-selected for the neutralization assays in Figure 4? Was this randomized?

Thank you. Capacity for live virus neutralisation is limited and it was not possible to assess the whole cohort. In order to include donors with a range of antibody responses participants were selected based on magnitude of Spike-specific response, 4 with the greatest response, 8 intermediate and 8 with low values. This has been added to the text.

10. Figure 4: it would be interesting to determine whether neutralization against a more neutralization-resistant variant (B.1.351 or B.1.617.2) is maintained in the elderly.

Yes, we agree. We are about to embark on further sampling of the whole cohort at a longitudinal time point (8 months since the booster vaccine) and intend to test the sera against variants of concern.

11. The correlation between age and the humoral and cellular response is difficult to interpret. Given the restricted age range, it can be expected that there will be limited differences, unless there are major differences in health between these individuals. The title of this paragraph should be tuned down and the restriction of this analysis should be mentioned in the Discussion section.

Thank you. We have changed the title of this section accordingly and have added to the Discussion regarding this limitation.

12. The Discussion section is somewhat limited. The authors should at least discuss the limitations of the study and indicate what additional information would be needed to draw firm conclusions.

Thank you for this advice. We have now added a paragraph to the Discussion to highlight the limitations of this study and also the requirement for longitudinal data on this cohort.

“One of the limitations to this study includes the lack of pre-vaccination sample which is a reflection of the speed at which the vaccination programme was rolled out in people over 80 years old and the challenge of operating within vaccine centres during national ‘lockdown’. Future work should assess the longevity of the observed responses and neutralisation of new variants that have emerged since the vaccination programme started. This is now of great interest and may help to guide the need for further booster doses. Our work has focused solely on donors aged 80 years and older and, as such, it will also be important to see how immunity compares in younger cohorts who receive the BNT162b2 vaccine on a 3 week interval”.

13. Please rephrase the sentences 74 to 80 in the introduction, which lacks proper understanding of immunological ageing.

Thank you. We agree that this was a poorly written section and has now been replaced.

14. What do the authors mean by independently-living people as mentioned in sentence 87 in the introduction? Is that the proper terminology? Is community dwelling more appropriate?

Thank you and we apologise for confusion. We have now elaborated on this in the text. These were individuals in community dwellings that did not require any assistance in daily living or care and were able to attend a vaccination hub.

15. Throughout the paper (including the title), the authors should avoid referring to the variants of concern with their country of origin. Either use the pango lineage names (P.1) or WHO's new Greek alphabet naming system for VOCs, in which P.1 is the γ variant.

Thank you. We have edited the text to contain the pango lineage throughout. We have left the Victoria designation as this is not country of origin but refers to first sequence prototype.